# Maternal mental-health treatment moderates the association between psychological distress and harsh parenting: A prospective cohort study

**Emily Midouhas**⬤*, **Bonamy R. Oliver**⬤

Department of Psychology and Human Development, UCL Institute of Education, University College London, London, United Kingdom

* emily.midouhas@ucl.ac.uk

## Abstract

### Objective

Parental psychological distress (depression, anxiety) is detrimental to child mental health. A key reason for this is that depressed and anxious parents are at risk of engaging in more negative, reactive and harsh parenting. While treatment for psychological distress has a long history of success in adults, less is known about how treatment for parental psychological distress may positively influence parenting behaviours, particularly in the general population. We examined the moderating role of mothers receiving treatment for depression or anxiety on the longitudinal relationship between maternal psychological distress and the development of harsh parenting (smacking and shouting) across early childhood (ages 3 to 7).

### Method

Using prospective data from 16,131 families participating in the UK's Millennium Cohort Study, we conducted moderator analysis within a multilevel repeated measures model to test whether receiving treatment for mental health problems could protect mothers with high psychological distress from engaging in harsh parenting.

### Results

In each wave, about 7% of mothers reported undergoing treatment for depression or anxiety at that time. Maternal psychological distress was associated with increased use of harsh parenting and that, adjusting for psychological distress, receiving psychological treatment was related to decreased use of harsh parenting. Importantly, receiving psychological treatment buffered the negative effect of psychological distress on harsh parenting.

### Conclusion

In early-to-middle childhood, mental health treatment may help mothers with depression or anxiety to be less harsh toward their children, thereby benefiting their child's psychological adjustment.

**Data Availability Statement:** All data are available from the UK Data Service (SN 5350, SN 5795, SN

6411, SN 7464) at beta.ukdataservice.ac.uk/
datacatalogue/series/series?id=2000031.

**Funding:** The author(s) received no specific
funding for this work.

**Competing interests:** The authors have declared
that no competing interests exist.

## Introduction

The importance of meeting the psychological needs of children and young people is increas-
ingly recognised [1]. Conveyors of biological (genetic, prenatal) risk as well as providers of
environmental context, parents have a crucial role in the onset and maintenance of children's
mental health and wellbeing and are seen as primary change agents [2]. Indeed, the role of
relationships between parents and children has been highlighted over many years [3]. In par-
ticular, a parenting environment characterised by sensitivity, consistency and warmth is seen
as protective for children's psychological adjustment [4, 5]. In contrast, a 'harsh' parenting
environment characterised by negative, hostile behaviors (e.g., smacking and shouting) is dis-
tinct from parental maltreatment [6], yet can have lasting detrimental impacts on children,
increasing the likelihood of internalising and externalising problems in the short- and long-
term [7]. As such, reducing experiences of harsh parenting is essential. This is often considered
to be best achieved through group-based parenting programmes [8], shown to benefit both
parent and child [9]. However, perhaps as much as one third of families do not benefit at all
[10] and effects on reducing parental stress and psychological distress seem to be relatively
transitory [11]. Finding other approaches to improving harsh parenting is crucial; one such
approach is to focus on the determinants of parental harshness.

Supported by empirical work in clinical and population samples over several decades, influ-
ential theoretical frameworks for understanding parenting behavior [12, 13] emphasize pri-
mary determinants of parenting under three broad categories: parental personal resources,
child characteristics, and contextual factors. Of contemporary interest under the category of
parental personal resources is psychological distress (e.g., symptoms of depression and anxi-
ety). A substantial proportion of parents report psychological distress [14], and this proportion
has increased in the pandemic era [15, 16]. There is plenty of evidence that maternal and pater-
nal psychological distress contributes substantially to child psychopathology [17, 18]. Yet less
is known about the mechanistic importance of father's psychological distress despite the fact
that fathers have a key role in the family system and an elevated risk for depression during the
transition to parenthood [18, 19]. For mothers, a mechanistic pathway linking distress to chil-
dren's psychological adjustment problems through poorer, harsher parenting is commonly
considered [20–22].

Albeit more generalizable to mothers than fathers given differences in parenting interven-
tion involvement, while evidence, including from a recent review [23] suggests that parental
psychological distress does not negatively impact parenting intervention effectiveness, success-
ful intervention programmes for children's mental health commonly have components that
aim to lower parental psychological distress as well as improve parenting behaviors [24]. Since
psychological distress in adults shows considerable treatment success outside of the parenting
context [25], it is plausible that such treatment also reduces harsh parenting, providing a cost-
effective means to improve outcomes for children also. This notion is empirically supported:
for example, a meta-analysis of the effectiveness of interventions aiming to prevent adverse
psychological outcomes in children of parents with mental illness suggested that there was "no
strong evidence supporting the hypothesis that interventions involving parents and children
were more beneficial than interventions aimed at parents only" [26, p.14].

Indeed, there is a small but growing literature from randomised-controlled trials suggesting
that successful treatment of parental psychological distress in turn improves parenting and
children's mental health, but across domains of parental mental health, there is some way to go
[27, 28]. Importantly, while these mechanisms are just as important to elucidate as potential
protective mechanisms to inform models of risk, in prospective general population samples,
less is known about the potential moderating role of receiving treatment for parental mental

health in the association between psychological distress and harsh parenting. Enhancing our understanding of these processes in this way is crucial since it tells us whether commonly received interventions for parental mental health translate into real-world improvements to harsh parenting and on to children's mental health.

We analysed data from the UK's Millennium Cohort Study (MCS), a large, prospective cohort study of families with young children, followed longitudinally from age 9 months. We explored longitudinal associations between maternal psychological distress and harsh parenting across early- and middle-childhood (ages 3, 5 and 7 years), as well as potential moderation of these associations as a function of mothers receiving treatment for depression or anxiety. While we acknowledge that the unique importance of paternal psychological distress and parenting on child outcomes is evidenced [18, 19] our study focused on mothers given the availability of data in the MCS.

In our general population sample, we hypothesised that children exposed to maternal psychological distress would have mothers who engage -- concurrently and longitudinally -- in more harsh parenting (i.e., more frequent smacking and shouting), relative to children without this exposure. We also expected to find that receiving treatment would moderate the link between psychological distress and harsh parenting, acting as a protective factor for those experiencing higher levels of psychological distress.

## Method

### Participants and procedure

MCS (www.cls.ioe.ac.uk/mcs) is a longitudinal survey drawing its sample from all UK births over one year, from 1 September 2000. The sample is disproportionately stratified to ensure adequate numbers in the four UK countries, electoral wards with disadvantaged and, in England, ethnic minority populations [29]. Ethical approval for MCS was gained from NHS Multi-Centre Ethics Committees; parents gave explicit written informed consent before interviews. Additional approval was not required for this secondary analysis. Sweeps 1–4 took place at child age 9 months, and 3, 5 and 7 years. We used records for one child per family (the first-born where there were twins or triplets), and all main analysis variables (parental psychological distress, treatment status and harsh parenting) were measured in Sweeps 2–4 (Fig 1 has a flow chart showing the data from waves 1–4 used in this study). Since growth curve modelling is able to handle unbalanced data, and to maximise power, our analytic sample comprised families with a score for harsh parenting in at least one of sweep (n = 16,131; 83.8% of MCS families).

### Measures

*Maternal Psychological Distress* was measured at children's ages 3, 5, and 7 years with the Kessler K6 [30], a 6-item screener with robust psychometric qualities ($\alpha$ = .87-.88). To aid interpretation of our sample and findings (see *Results*), we also categorised high and low levels of psychological distress, defined as high (1 standard deviation above the mean or higher; at least a score of 6) and low (1 standard deviation below the mean or lower; a score of 0).

*Harsh parenting* was assessed with two items measuring how often the parent smacks and shouts when the child misbehaves on a five-point Likert scale from 1 = 'never' to 5 = 'daily'. These two questions pertain to the dimensions of corporeal punishment/physical assault and psychological aggression on the Parent–Child Conflict Tactics Scales [31] and have been used as indicators of the harsh parenting construct in cohort studies elsewhere [32, 33]. Items were summed to generate a total score, whereby higher values indicated more frequent use of these tactics ($\alpha$ = .52-.54 across sweeps).

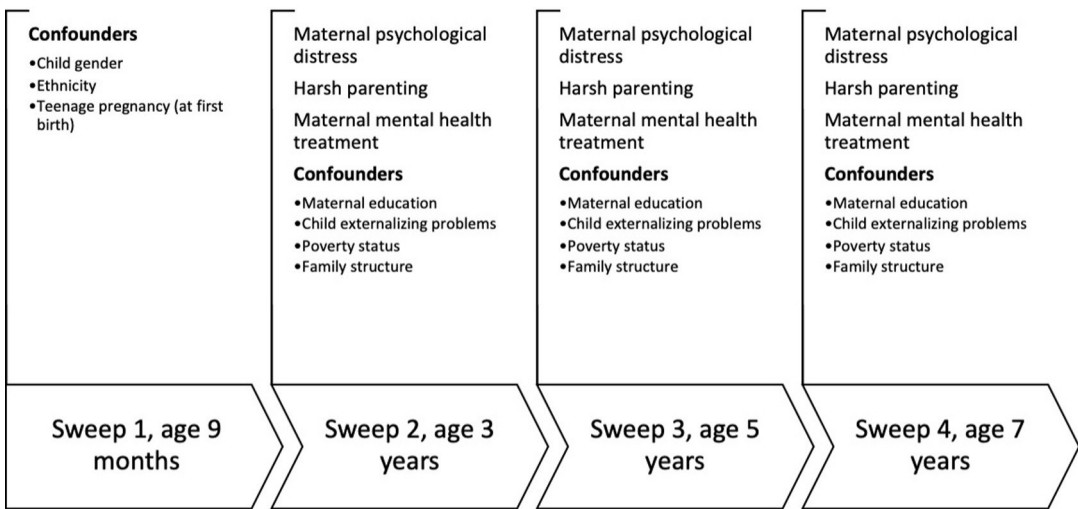

**Fig 1. Flow chart of MCS waves with data used in this analysis.**

**Potential confounders.** We adjusted for key child, parent and family characteristics in order to factor out confounders of the association between maternal psychological distress and harsh parenting. Key child and parent covariates were child gender, ethnicity, child externalising problems, maternal education, poverty status, teenage pregnancy (at first birth) and family structure. Externalising problems were measured using parent-reported conduct problems and hyperactivity/inattention scales of the Strengths and Difficulties Questionnaire (SDQ [34]; α = .78 to .80). With regard to family-level covariates, income poverty status (below the poverty line, set for equivalised net family income at 60% of the UK national median household income) and family structure (two parents or not) were time-varying, measured at ages 3, 5 and 7 years. Maternal education was measured with a binary indicator of whether the mother achieved a university degree or higher degree by the end of our study period (age 7 years).

## Data preparation and analytic strategy

First, we investigated whether the MCS families in our analytic sample (n = 16,131) were significantly different on our study variables (at $p < .05$) from those not (n = 3,113). We then explored levels and patterns of missingness in our covariates to decide on our approach to dealing with missing data. Following this, we inspected the correlations between our main variables. Finally, we examined the relationships between maternal psychological distress, depression/anxiety treatment status and harsh parenting across early-to-middle childhood by fitting 2-level growth curve models [35] where occasions of harsh parenting measurements (Level 1) were nested in children (Level 2). These models allowed us to estimate the average level of harsh parenting at a particular time-point as well as the average rate of change in harsh parenting over time. We fitted both fixed and random linear slopes. By specifying a random linear slope on the child's age to allow for changes in harsh parenting across time to vary between children, we were also able to model individual trajectories of harsh parenting from ages 3 to 7. The stratified sampling design of MCS was recognised by including the nine MCS strata in all models: England-advantaged, England-disadvantaged, England-ethnic, Wales-advantaged, Wales-disadvantaged, Scotland-advantaged, Scotland-disadvantaged, Northern Ireland-advantaged and Northern Ireland-disadvantaged. All descriptive estimates are weighted using the svy commands in Stata (but Ns are unweighted).

We estimated three models. Model 1 included age (grand mean centred at age 5.22 years). Also included were maternal psychological distress and treatment status, as well as the interaction terms for psychological distress*age and treatment status*age. This model enabled us to examine whether levels of harsh parenting at around age 5 and rate of change in harsh parenting over time shifted with psychological distress and treatment status. Model 2 included the interaction for psychological distress*treatment status, also specified as an interactive effect with age. This model therefore tested whether treatment status moderated the association between psychological distress and harsh parenting. Model 3, adding the child and family covariates, tested the robustness of all effects identified.

## Results

### Sample bias analysis

First, we conducted a bias analysis by examining the characteristics of our analytic sample (n = 16,131) compared to those in the rest of the MCS sample (n = 3,113). We included only singleton children and first-born twins or triplets in both samples. The descriptives of the full set of variables included in the regression models for these two samples are found in Tables 1 and 2. Statistically significant differences were found across all demographic variables and treatment status but they were not found for harsh parenting or psychological distress. The analytic sample comprised more socio-economically advantaged families and they were more likely to report being treated for mental health problems.

### Correlations

All correlations between main variables were statistically significant (Table 3). Moderate correlations were found between PD at ages 3, 5 and 7 and between HP at ages 3, 5 and 7, as expected. Weak correlations were shown between PD and HD, ranging .11-.16 across ages.

### Growth curve models

All model results are shown in Table 4. We found evidence of moderate stability in harsh parenting scores over time, as indicated by the intraclass correlation coefficients (ICCs; proportions of within-parent variation to total variation), ranging from .49 to .52.

Model 1 showed psychological distress to be significantly related to harsh parenting at age 5, but not to the linear rate of change in harsh parenting. Adjusting for psychological distress, receiving mental health treatment for depression or anxiety was significantly associated with lower harsh parenting at age 5 and negatively related to the linear rate of change in harsh parenting, such that mothers receiving treatment had an annual decrease in their harsh parenting over time relative to mothers not receiving treatment.

Model 2 showed evidence of moderation by treatment for depression and anxiety: the effect of psychological distress on harsh parenting was weaker at central age in children whose mothers received treatment ($b = 0.020$, SE = 0.006, $p < .01$). However, there was no such interaction effect on the linear rate of change in harsh parenting over time.

Adding our key covariates in Model 3 showed that mothers of girls and Indian and Pakistani children (compared to White) demonstrated lower harsh parenting but mothers of Black children showed more harsh parenting. Non-teen mothers and those who were non-poor, university-educated and from intact families had lower harsh parenting. Mothers of children with higher externalising problems showed increased harsh parenting. Importantly, adding these child and family covariates did not attenuate the main effect of psychological distress or its interaction with receiving treatment.

**Table 1. Descriptives of categorical study variables in the analytic and non-analytic samples.**

| Variable | *Analytic sample* (*n* = 16,131) | | *Non-analytic sample* (n = 3113) | | *Test* |
| --- | --- | --- | --- | --- | --- |
| | *N* | *%* | *N* | *%* | *F* |
| | | | **Child** | | |
| Girl | 7,887 | 48.92 | 1462 | 46.62 | 3.86* |
| Ethnicity | | | | | 101.73*** |
| White | 13,735 | 89.31 | 2006 | 74.25 | |
| Black | 477 | 3.05 | 117 | 4.00 | |
| Indian | 389 | 1.69 | 108 | 2.46 | |
| Pakistani or Bangladeshi | 829 | 2.88 | 521 | 10.38 | |
| Mixed | 506 | 2.19 | 223 | 5.49 | |
| Other | 187 | 0.88 | 116 | 3.42 | |
| | | | **Parent or household** | | |
| Depression/anxiety treatment status | | | | | |
| Age 3 | 1251 | 7.21 | 64 | 1.81 | 99.22*** |
| Age 5 | 1247 | 7.08 | 22 | 0.44 | 159.84*** |
| Age 7 | 1247 | 7.49 | 53 | 1.66 | 119.65*** |
| Mother is university-educated | 2770 | 20.36 | 256 | 10.64 | 89.78*** |
| Teenage mother status (first birth) | 3352 | 17.64 | 811 | 24.36 | 58.39*** |
| Two-parent family | | | | | |
| Age 3 | 12,985 | 87.05 | 2311 | 77.96 | 121.80*** |
| Age 5 | 11,775 | 82.43 | 559 | 72.04 | 34.17*** |
| Age 7 | 11,080 | 77.07 | 364 | 71.01 | 4.05* |
| Poverty status | | | | | |
| Age 3 | 3830 | 24.72 | 291 | 56.89 | 154.89*** |
| Age 5 | 4331 | 26.59 | 229 | 73.80 | 168.30*** |
| Age 7 | 3798 | 22.34 | 364 | 53.61 | 121.87*** |

*Note*: *$p < .05$,

**$p < .01$,

***$p < .001$. F = F statistic for design-based Pearson chi-square (converted to F test to account for the MCS sampling design). Proportions are weighted to account for sampling design and non-response in MCS. Ns are unweighted.

To illuminate the interaction between psychological distress and receiving treatment, we plotted the predicted trajectories of harsh parenting for illustrative cases of mothers with high levels of psychological distress by treated vs. untreated status and for the mother with a low level of psychological distress (Fig 2). Predicted values were plotted for the averages of continuous covariates and the reference groups for categorical covariates. Fig 2 suggests that mothers with high PD who are treated show a reduction in harsh parenting. Specifically, the predicted trajectories for mothers high in PD (treated) and high in PD (untreated) start (age 3) with higher harsh parenting scores than those with low PD. However, mothers with high PD who undergo treatment reduce in harsh parenting across the period to align with mothers with low PD at age 7. Mothers low in PD show a flat trajectory across ages 3 to 7 with a slight steady increase. Conversely, mothers with high PD who are untreated remain at a higher level of PD across this period.

## Discussion

Despite our understanding that negative, including harsh, parenting is one pathway through which psychological distress can lead to poor behavioral outcomes in children and that there is

**Table 2. Descriptives of continuous time-varying study variables in the analytic and non-analytic samples.**

| Variable | Analytic sample (n = 16,131) | | | | Non-analytic sample (n = 3,113) | | | |
|---|---|---|---|---|---|---|---|---|
| | N | M(SE) | Range | 95% CI | N | M(SE) | Range | 95% CI |
| HP | | | | | | | | |
| Age 3 | 13.121 | 5.28(0.03) | 2–10 | [5.23,5.32] | 29 | 4.84(0.43) | 2–9 | [3.99,5.68] |
| Age 5 | 14,190 | 4.78(0.02) | 2–10 | [4.75,4.81] | 7 | 4.79(1.06) | 2–9 | [2.71,6.87] |
| Age 7 | 12,502 | 4.65(0.02) | 2–10 | [4.62,4.68] | 1* | - | - | - |
| PD | | | | | | | | |
| Age 3 | 12,381 | 2.87(0.03) | 0–24 | [2.81,2.94] | 76 | 2.70(0.37) | 0–14 | [1.98,3.43] |
| Age 5 | 13,774 | 2.77(0.04) | 0–24 | [2.70,2.84] | 5 | 6.30(2.42) | 0–12 | [1.54,11.05] |
| Age 7 | 13,607 | 2.31(0.03) | 0–20 | [2.25,2.37] | 1* | - | - | - |
| Child's age (years) | | | | | | | | |
| Age 3 | 14,796 | 3.13(0.003) | 2.65–4.57 | [3.12,3.14] | 785 | 3.21(0.01) | 2.90–4.46 | [3.18,3.24] |
| Age 5 | 14,760 | 5.21(0.004) | 4.41–6.13 | [5.20,5.22] | 484 | 5.21(0.02) | 4.44–6.05 | [5.18,5.24] |
| Age 7 | 13,227 | 7.23(0.01) | 6.34–8.15 | [7.22,7.24] | 580 | 7.29(0.01) | 7.27–7.32 | [1.38,1.75] |
| Child externalising problems | | | | | | | | |
| Age 3 | 14,170 | 6.58(0.06) | 0–20 | [6.47,6.70] | 424 | 7.41(0.24) | 0–19 | [6.95,7.89] |
| Age 5 | 14,422 | 4.67(0.05) | 0–20 | [4.59,4.76] | 226 | 5.87(0.27) | 0–17 | [5.34,6.41] |
| Age 7 | 13,007 | 4.62(0.05) | 0–20 | [4.52,4.72] | 406 | 5.47(0.27) | 0–18 | [4.94,6.01] |

*Note*: All means are weighted to account for both sampling design and non-response in MCS. Ns are unweighted. CI = Confidence Interval. Standard errors and CIs are adjusted for clustered sampling except in cases marked

* based on a sample represented by a single primary sampling unit (i.e., ward) for a given sampling stratum. These discrepancies arise occasionally because of migration out of the original strata. HP = Harsh parenting; PD = Psychological distress

good evidence for the treatment for parental psychological distress [20] we know little about how intervention for parental psychological distress may benefit parenting behaviors, especially prospectively in the general population. This is the first study, to our knowledge, that explores the moderating effect of treatment in the relationship between maternal psychological distress and harsh parenting in a prospective, longitudinal general population study. Understanding how the impact of maternal psychological distress on harsh parenting can be mitigated is crucial to inform models of risk and resilience in relation to both parental and child mental health. In addition, if treatment for parents reduces harsh parenting, the likely knock-on effects for children's mental health could help to maximise stretched mental-health services.

Our findings from a prospective UK cohort study across early to mid-childhood showed that, when considered together, maternal psychological distress and treatment status were

**Table 3. Correlations among the main variables in the analytic sample (n = 16,131).**

| | PD age 3 | PD age 5 | PD age 7 | HP age 3 | HP age 5 |
|---|---|---|---|---|---|
| PD age 3 | 1 | | | | |
| PD age 5 | .54*** | 1 | | | |
| PD age 7 | .51*** | .55*** | 1 | | |
| HP age 3 | .16*** | .11*** | .12*** | 1 | |
| HP age 5 | .12*** | .14*** | .12*** | .56*** | 1 |
| HP age 7 | .11*** | .12*** | .16*** | .51*** | .62*** |

*Note*:***p < .001*. 3, 5 and 7 refer to age in years. PD = Psychological distress;
HP = Harsh parenting

**Table 4. Fixed effects estimates and variance covariance estimates for Models 1–3 predicting harsh parenting.**

| | Model 1 (n = 15,728) | | | Model 2 (n = 15,728) | | | Model 3 (n = 15,261) | | |
|---|---|---|---|---|---|---|---|---|---|
| | Coeff. | SE | 95% CI | Coeff. | SE | 95% CI | Coeff. | SE | 95% CI |
| | | | | | Fixed effects | | | | |
| Constant | 4.721*** | 0.202 | [4.681,4.761] | 4.707*** | 0.021 | [4.666,4.747] | 4.116*** | 0.029 | [4.058,4.174] |
| Age | 0.153*** | 0.003 | [0.090,0.254] | 0.168*** | 0.043 | [0.083,0.253] | 0.068 | 0.043 | [-0.015,0.153] |
| PD | 0.071*** | 0.003 | [0.066,0.076] | 0.078*** | 0.003 | [0.071,0.083] | 0.045*** | 0.003 | [0.039,0.051] |
| PD*age | -0.008 | 0.010 | [-0.028,0.012] | -0.006 | 0.011 | [-0.028,0.016] | -0.011 | 0.011 | [-0.034,0.011] |
| Treatment status | -0.139*** | 0.027 | [-0.192,-0.085] | 0.040 | 0.044 | [-0.046,0.127] | -0.043 | 0.043 | [-0.129,0.042] |
| Treatment status*age | -0.227* | 0.106 | [-0.435,-0.019] | -0.181 | 0.180 | [-0.535,0.173] | -0.081 | 0.179 | [-0.432,0.269] |
| PD*treatment status | | | | -0.033*** | 0.006 | [-0.046,-0.021] | -0.020** | 0.006 | [-0.033,-0.008] |
| PD*treatment status*age | | | | -0.009 | 0.026 | [-0.060,0.043] | 0.0002 | 0.026 | [-0.051,0.051] |
| Female child | | | | | | | -0.088*** | 0.019 | [-0.126,0.049] |
| *Child ethnicity (Ref: White)* | | | | | | | | | |
| Mixed | | | | | | | 0.049 | 0.060 | [-0.069,0.167] |
| Indian | | | | | | | -0.186* | 0.074 | [-0.331,-0.041] |
| Pakistani or Bangladeshi | | | | | | | -0.252*** | 0.060 | [-0.370,-0.135] |
| Black | | | | | | | 0.145* | 0.066 | [0.015,0.274] |
| Other | | | | | | | -0.096 | 0.102 | [-0.296,0.105] |
| Child externalising problems | | | | | | | 0.114*** | 0.002 | [0.129, 0.138] |
| Teenage mother | | | | | | | -0.269*** | 0.027 | [-0.321,-0.217] |
| Mother is university-educated | | | | | | | -0.095*** | 0.026 | [-0.147,-0.043] |
| Two-parent family | | | | | | | 0.174*** | 0.021 | [0.132,0.216] |
| Family poverty status | | | | | | | -0.075*** | 0.019 | [-0.111,-0.038] |
| *Area stratum (Ref: England-advantaged)* | | | | | | | | | |
| England-disadvantaged | -0.005 | 0.027 | [-0.607,0.050] | -0.005 | 0.028 | [-0.061,0.050] | -0.031 | 0.026 | [-0.084,0.023] |
| England-ethnic | -0.224*** | 0.039 | [-0.310,-0.157] | -0.238*** | 0.039 | [-0.314,-0.161] | -0.111*** | 0.048 | [-0.205,-0.018] |
| Scotland-advantaged | 0.013 | 0.045 | [-0.076,0.101] | 0.014 | 0.045 | [-0.075,0.102] | 0.051 | 0.043 | [-0.032,0.135] |
| Scotland-disadvantaged | -0.019 | 0.046 | [-0.109,0.071] | -0.017 | 0.046 | [-0.107,0.072] | -0.007 | 0.044 | [-0.092,0.078] |
| Northern Ireland-advantaged | 0.275*** | 0.055 | [0.168,0.382] | 0.276*** | 0.055 | [0.169,0.383] | 0.337*** | 0.052 | [0.236,0.439] |
| Northern Ireland-disadvantaged | 0.031 | 0.046 | [-0.059,0.121] | 0.035 | 0.046 | [-0.055,0.125] | 0.076 | 0.045 | [-0.011,0.163] |
| Wales-advantaged | -0.036 | 0.048 | [-0.137,0.064] | -0.035 | 0.051 | [-0.136,0.065] | -0.033 | 0.048 | [-0.128,0.062] |
| Wales-disadvantaged | -0.068 | 0.037 | [-0.141,0.005] | -0.067 | 0.037 | [-0.140,0.006] | -0.093 | 0.036 | [-0.163,-0.024] |
| | | | | | Random effects | | | | |
| Level 2 (child) | | | | | | | | | |
| Between-child intercept variance | 1.154*** | 0.019 | [1.116,1.192] | 1.151*** | 0.019 | [1.114,1.190] | 0.096*** | 0.017 | [0.927,0.994] |
| Between-child slope variance | 0.483*** | 0.126 | [0.289,0.806] | 0.483*** | 0.126 | [0.289,0.806] | 0.336*** | 0.119 | [0.168,0.672] |
| Between-child intercept/slope covariance | 0.175*** | 0.040 | [0.097, 0.252] | 0.174*** | 0.040 | [0.096,0.252] | 0.158*** | 0.036 | [0.089, 0.228] |
| Level 1 (occasion) | | | | | | | | | |
| Between-occasion variance | 1.067*** | 0.011 | [1.045,1.089] | 1.067*** | 0.011 | [1.045,1.089] | 0.968*** | 0.011 | [0.947,0.989] |
| Intra-class correlation | .520 | 0.005 | [0.509, 0.530] | .519 | 0.005 | [0.508, 0.530] | .498 | 0.006 | [0.487,0.509] |

*Note*: *$p < .05$,

**$p < .01$,

***$p < .001$. PD = psychological distress

related to an increase and a decrease, respectively, in the level of harsh parenting (smacking and shouting) directed towards their children. A large body of literature shows links between psychological distress in mothers and fathers and parenting behaviors [17, 18]. Of particular

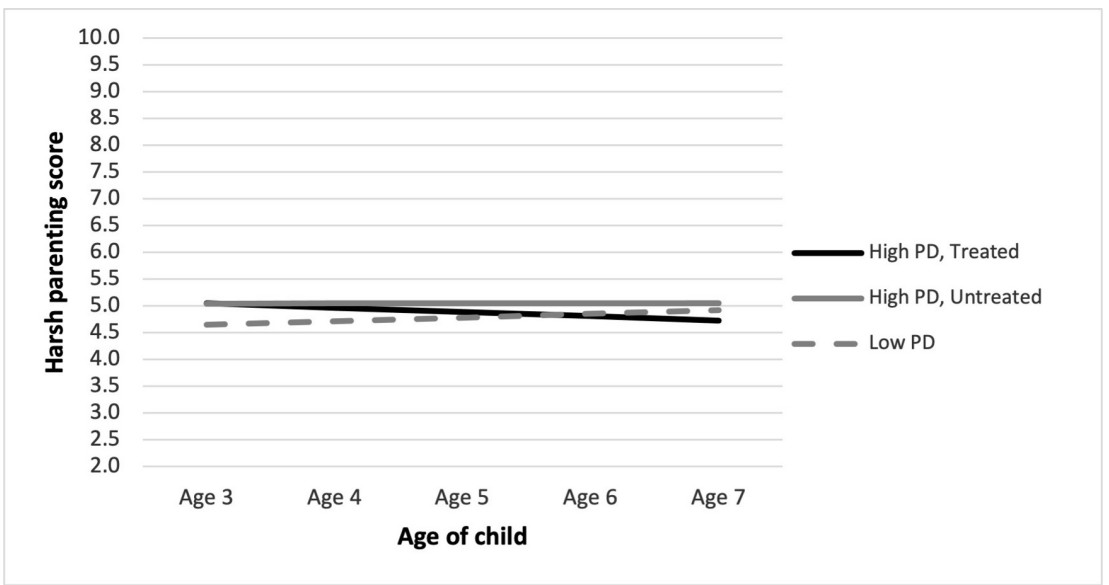

**Fig 2. Predicted harsh parenting trajectories by high/low psychological distress and treatment status (Fully-adjusted Model 3).** *Note*: Predictions are plotted for the reference group for each categorical variable and at the mean of each continuous variable.

interest was that, as hypothesised, mothers with higher psychological distress who were undergoing treatment for depression or anxiety demonstrated lower harsh parenting than those not undergoing treatment at the time. Therefore, we suggest that treatment acted as a protective factor for harsh parenting since it was associated with lower- than-expected levels of harsh parenting, despite self-reported higher psychological distress. This means that depressed or anxious mothers may be better able to "manage" their parenting behaviors–likely through increased regulation and decreased reactivity [36]—despite experiencing these mental health problems if they were receiving support for them. Although we do not know the type of treatment mothers were receiving (likely one of the first-line treatments of talk or cognitive behavioral therapy and selective serotonin reuptake inhibitor [SSRI] pharmacotherapy [37]), it appears that receiving some form of support for mental health problems can help to break the link between psychological distress and harsh parenting behavior.

Despite the strengths of our study including the sample size, and longitudinal, prospective nature of the data available, it had additional limitations. First, as a correlational study we are unable to prove that maternal psychological distress caused mothers to parent more harshly, or, similarly, that undergoing treatment itself caused mothers to be less harsh. Second, related to this, although we can speculate, we do not know the reasons why treatment was beneficial in this way. Third, as previously mentioned, we did not know the type of treatment mothers had, including whether this was medication (e.g., antidepressants or another psychiatric drug) and, if so, what dose, therapy, another form of treatment, or a combination of these, as this information was not sought from parents in the MCS. Future research should aim to explore the differential effects of treatment types for parenting and child outcomes through mental health services data linkage to general population cohorts, if possible. Fourth, although we know that those receiving treatment had received a diagnosis of depression or severe anxiety, we do not know the exact diagnoses or the severity of presentation of symptoms. Fifth, the use of maternal reports to measure both psychological distress and harsh discipline might inflate correlations between these measures, and both of these maternal indicators could be subject to biases related to social desirability. Sixth, we did not have data on fathers' harsh parenting as

these were unavailable in the MCS. Future research should attempt to examine how treatment for mental health problems could help fathers to reduce their harsh parenting, alongside mothers, given increasing research on the importance of fathers' psychological distress and parenting for child outcomes [38]. Further exploration of the momentary interactions between children and parents (fathers and mothers) who have depression or anxiety could also help us uncover the mechanisms driving these relationships [39]. Finally, another avenue we suggest for future research, beyond the scope of the current study, is to consider key relevant confounding variables such as emotion regulation.

Our findings have considerable implications for improving overall family wellbeing in the community. Treating parents' mental health problems may–at least for some families—lessen parenting practices that are detrimental to children's mental health, reducing the need for parenting intervention, and, arguably, intervention for children's mental health difficulties down the line. With regard to implications for the clinical context, our findings suggest that when a child is referred to mental health services, screening for parents' psychological distress would be prudent to determine whether mental health support might also be needed for the child's parents, alongside or even instead of parenting intervention and other forms of treatment where parents are seen as the agents of change. We emphasise that there is need to replicate this work in other community samples, as well as to unpick the specifics of treatment to better understand these processes, but the potential for maximising already stretched mental-health services suggested by our findings is nevertheless attractive.

## Acknowledgments

The authors thank the Millennium Cohort Study families for their time and participation, as well as the Centre for Longitudinal Studies (CLS) and the UK Data Service for access.

## Author Contributions

**Conceptualization:** Emily Midouhas, Bonamy R. Oliver.

**Data curation:** Emily Midouhas.

**Formal analysis:** Emily Midouhas.

**Methodology:** Emily Midouhas, Bonamy R. Oliver.

**Writing – original draft:** Emily Midouhas, Bonamy R. Oliver.

**Writing – review & editing:** Emily Midouhas, Bonamy R. Oliver.

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
