## [Decision Letter · Decision Letter 0]

25 Jan 2023

PONE-D-22-21569Maternal mental-health treatment moderates the association between psychological distress and harsh parenting: A prospective cohort studyPLOS ONE

Dear Dr. Midouhas,

Thank you for submitting your manuscript to PLOS ONE. After careful consideration, we feel that it has merit but does not fully meet PLOS ONE’s publication criteria as it currently stands. Therefore, we invite you to submit a revised version of the manuscript that addresses the points raised during the review process.

ACADEMIC EDITOR: Thank you for considering PLOS ONE for your paper. I read your study with interest, and based on the reviewers' reports, minor revisions are suggested

We look forward to receiving your revised manuscript.

Kind regards,

Giulia Ballarotto

Academic Editor

PLOS ONE

Journal Requirements:

Additional Editor Comments (if provided):

Thank you for considering PLOS ONE for your paper. I read your study with interest, and based on the reviewers' reports, minor revisions are suggested

Reviewers' comments:

Reviewer's Responses to Questions

**Comments to the Author**

1. Is the manuscript technically sound, and do the data support the conclusions?

Reviewer #1: Yes

Reviewer #2: Yes

2. Has the statistical analysis been performed appropriately and rigorously? 

Reviewer #1: Yes

Reviewer #2: Yes

3. Have the authors made all data underlying the findings in their manuscript fully available?

Reviewer #1: Yes

Reviewer #2: Yes

4. Is the manuscript presented in an intelligible fashion and written in standard English?

Reviewer #1: Yes

Reviewer #2: Yes

5. Review Comments to the Author

Reviewer #1: This is an interesting study addressing an important issue. The manuscript is clear to read and presents some strenghts as the sample size and the longitudinal nature. I have only some minor comments:

-I suggest to insert in the limits that other relevant variables, as affect regulation capabilities, should be explored in future studies since may have a role as confounder.

-the study has a longitudinal nature with several waves, probabli a flow-chart woul be usefull.

Minor:in acknolegment section the first sentence regards more authors but I think there is only an author in this manuscript.

Reviewer #2: Thank you very much for the possibility to review the study “Maternal mental-health treatment moderates the association between psychological distress and harsh parenting: A prospective cohort study”. The study aimed to examine the moderating role of mothers receiving treatment for depression or anxiety on the longitudinal relationship between maternal psychological distress and the development of harsh parenting across early childhood. This longitudinal study is very interesting and well written.

I suggest only a few minor revisions:

1) as much as the authors have included that one of the limitations is the absence of data on fathers, I think it may be important that the paternal role is nevertheless also considered in the introduction to the work. Several studies have investigated the role of fathers and especially the mother-child and father-child interactive quality, so it would be important to be able to explore this literature more, then highlight the specific focus of the paper on mothers. For example, there are several recent studies:

- Cabrera, N. J., Volling, B. L., & Barr, R. (2018). Fathers are parents, too! Widening the lens on parenting for children's development. Child Development Perspectives, 12(3), 152-157.

- Barker, B., Iles, J. E., & Ramchandani, P. G. (2017). Fathers, fathering and child psychopathology. Current opinion in psychology, 15, 87-92.

- Cimino, S., Cerniglia, L., Tambelli, R., Ballarotto, G., Erriu, M., Paciello, M., ... & Koren-Karie, N. (2020). Dialogues about emotional events between mothers with anxiety, depression, anorexia nervosa, and no diagnosis and their children. Parenting, 20(1), 69-82.

2) it would be useful to include a figure to represent the different waves of the longitudinal study.

3) Lastly, it is suggested that the clinical implications of the study's findings should be more fully explored

6. PLOS authors have the option to publish the peer review history of their article (what does this mean?). If published, this will include your full peer review and any attached files.

Reviewer #1: No

Reviewer #2: No

---

## [Author Response · Author response to Decision Letter 0]

7 Feb 2023

We thank you as Academic Editor and the reviewers for the opportunity to revise our manuscript, ‘Maternal mental-health treatment moderates associations between psychological distress and harsh parenting: evidence from a UK population sample across childhood’ (PONE-D-22-21569) for publication in PLOS ONE. We very much appreciate acknowledgements of the study strengths and for comments that we have found to be constructive and helpful for improving the paper.

Below, we detail our responses, point-by-point and highlight where changes are made to the manuscript for ease of reference. We should also note that we have added a middle initial for the second author which we missed in our submitted manuscript.

Academic Editor

Thank you for considering PLOS ONE for your paper. I read your study with interest, and based on the reviewers' reports, minor revisions are suggested.

Response: Thank you for your encouragement of our work!

Reviewer 1:

Comment: This is an interesting study addressing an important issue. The manuscript is clear to read and presents some strenghts as the sample size and the longitudinal nature.

Response: Thank you!

Comment: I suggest to insert in the limits that other relevant variables, as affect regulation capabilities, should be explored in future studies since may have a role as confounder.

Response: Thank you for this important point. This is now included in limitations:

Another avenue we suggest for future research, beyond the scope of the current study, is to consider key relevant confounding variables such as emotion regulation. 

Comment: the study has a longitudinal nature with several waves, probably a flow-chart would be useful.

Response: Thank you, we hadn’t considered a flowchart, but it’s a good idea! This is now included in the Method section. 

Comment: Minor:in acknowledgment section the first sentence regards more authors but I think there is only an author in this manuscript.

Response: There are two authors of this paper which is why we use ‘authors’ rather than ‘author’. 

Reviewer 2:

Thank you very much for the possibility to review the study “Maternal mental-health treatment moderates the association between psychological distress and harsh parenting: A prospective cohort study”. The study aimed to examine the moderating role of mothers receiving treatment for depression or anxiety on the longitudinal relationship between maternal psychological distress and the development of harsh parenting across early childhood. This longitudinal study is very interesting and well written.

Response: Thank you for your kind review.

Comment: 1) as much as the authors have included that one of the limitations is the absence of data on fathers, I think it may be important that the paternal role is nevertheless also considered in the introduction to the work. Several studies have investigated the role of fathers and especially the mother-child and father-child interactive quality, so it would be important to be able to explore this literature more, then highlight the specific focus of the paper on mothers. For example, there are several recent studies:

- Cabrera, N. J., Volling, B. L., & Barr, R. (2018). Fathers are parents, too! Widening the lens on parenting for children's development. Child Development Perspectives, 12(3), 152-157.

- Barker, B., Iles, J. E., & Ramchandani, P. G. (2017). Fathers, fathering and child psychopathology. Current opinion in psychology, 15, 87-92.

- Cimino, S., Cerniglia, L., Tambelli, R., Ballarotto, G., Erriu, M., Paciello, M., ... & Koren-Karie, N. (2020). Dialogues about emotional events between mothers with anxiety, depression, anorexia nervosa, and no diagnosis and their children. Parenting, 20(1), 69-82.

Response: Thank you – this is an excellent point. Although we acknowledged this as a limitation in our Discussion, it is indeed important to discuss fathers earlier in the paper. We now include points about fathers in the introduction to provide this background (citing the above papers) and go on to explain at the end of the introduction why we focused on mothers in this analysis. 

Comment: 2) it would be useful to include a figure to represent the different waves of the longitudinal study.

Response: Thank you. As mentioned above, we now include this in the Method section.

Comment: 3) Lastly, it is suggested that the clinical implications of the study's findings should be more fully explored.

Response: We have now extended our discussion of implications to cover clinical implications (see final paragraph of Discussion).

---

## [Editor Report · Decision Letter 1]

8 Feb 2023

Maternal mental-health treatment moderates the association between psychological distress and harsh parenting: A prospective cohort study

PONE-D-22-21569R1

Dear Dr. Midouhas,

We’re pleased to inform you that your manuscript has been judged scientifically suitable for publication and will be formally accepted for publication once it meets all outstanding technical requirements.

Kind regards,

Giulia Ballarotto

Academic Editor

PLOS ONE
---

## [Editor Report · Acceptance letter]

14 Feb 2023

PONE-D-22-21569R1 

Maternal mental-health treatment moderates the association between psychological distress and harsh parenting: A prospective cohort study 

Dear Dr. Midouhas:

I'm pleased to inform you that your manuscript has been deemed suitable for publication in PLOS ONE. Congratulations! Your manuscript is now with our production department. 

Kind regards, 

on behalf of

Dr Giulia Ballarotto 

Academic Editor

PLOS ONE